# Risk Stratification of Local Flaps and Skin Grafting in Skin Cancer-Related Facial Reconstruction: A Retrospective Single-Center Study of 607 Patients

**DOI:** 10.3390/jpm12122067

**Published:** 2022-12-15

**Authors:** Frederik Penzien Wainer Mamsen, Claes Hannibal Kiilerich, Jørgen Hesselfeldt-Nielsen, Iselin Saltvig, Celine Lund-Nielsen Remvig, Hannah Trøstrup, Volker-Jürgen Schmidt

**Affiliations:** 1Department of Plastic Surgery and Breast Surgery, Zealand University Hospital (SUH), University of Copenhagen, 4000 Roskilde, Denmark; 2Department of Clinical Medicine, University of Copenhagen, 2200 Copenhagen, Denmark

**Keywords:** non-melanoma skin cancer, facial reconstruction, surgical options, surgical complications

## Abstract

**Background:** Non-melanoma skin cancer (NMSC) takes up a substantial fraction of dermatological and plastic surgical outpatient visits and surgeries. NMSC develops as an accumulated exposure to UV light with the face most frequently diagnosed. **Method:** This retrospective study investigated the risk of complications in relation to full-thickness skin grafts (FTSG) or local flaps in 607 patients who underwent facial surgery and reconstruction at a high-volume center for facial cancer surgery at a tertiary university hospital. **Results:** Between 01.12.2017 and 30.11.2020, 304 patients received reconstructive flap surgery and 303 received FTSG following skin cancer removal in the face. Flap reconstruction was predominantly performed in the nasal region (78%, *n* = 237), whereas FTSG reconstruction was performed in the nasal (41,6%, *n* = 126), frontal (19.8%, *n* = 60), and temporal areas (19.8%, *n* = 60), respectively. Patients undergoing FTSGs had a significantly higher risk of hematoma (*p* = 0.003), partial necroses (*p* < 0.001), and total necroses (*p* < 0.001) compared to flap reconstruction. Age and sex increased the risk of major complications (hematoma, partial or total necrosis, wound dehiscence, or infection) for FTSG, revealing that men exhibited 3.72 times increased risk of major complications compared to women reconstructed with FTSG. A tumor size above 15 mm increased the risk of hematoma and necrosis significantly. In summary, local flaps for facial reconstruction after skin cancer provide lower complication rate compared with FTSGs, especially in elderly and/or male patients. The indication for FTSG should be considered critically if the patient’s tumor size and location allow for both procedures.

## 1. Introduction

Skin cancer diagnoses are frequent and predominantly affect light-skinned individuals. It is the leading type of cancer globally [1,2] and takes up around 50% of all diagnoses in dermatological outpatient clinics [3]. Malignant skin cancer divides into melanoma- and non-melanoma skin cancer (NMSC), with NMSC being the most frequent, especially in the head and neck region of the elderly. NMSC comprises multiple types but predominantly consists of basal cell carcinomas (BCC) and squamous cell carcinomas (SCC) [2]. 

Facial cancer surgery and reconstruction pose a significant surgical challenge. The four pillars of reconstructive oncology, namely, clearance, form, function, and patient satisfaction, must be considered to ensure the tumor removal with consideration of the aesthetics subunits [4]. The decision on which surgical reconstructive technique to use is challenging but can be mastered by a skilled and knowledgeable surgeon. Often, simple suture is preferred if the tumor is small or placed in an easy-to-excise location. For tumors deemed too large for simple suture either due to location, tumor size, or aesthetic outcome, the defect can be reconstructed using full-thickness skin graft (FTSG), local flap, split-thickness skin grafts, or distant flaps. The decision is not clear-cut but includes patient health and comorbidities, tumor location and size, published data based on surgical empiricism, etc. [2,5,6]. The surgeon must therefore consider every surgical treatment to ensure the lowest risk of complications with regard to the patient’s wishes for a satisfying outcome [4].

This present study compares FTSGs and local flaps as treatment options in terms of complications after surgical removal of facial NMSC. By dividing the face into seven regions, the study further investigates surgical preferences according to distinct facial regions and the impact of patient demographics on complications. Additionally, we propose data to guide the surgeon in choosing and managing the best reconstructive technique in the complex field of facial reconstruction.

## 2. Materials and Methods

The retrospective quality assurance and optimization study is approved by the local authorities (approval number REG-098-2020). The study complies with the patient data protection regulation in Denmark. Data was collected between 01.12.2021 and 25.04.2022 and included patients undergoing facial surgery for BCC and SCC from 01.12.2017 to 01.12.2020 at the Department of Plastic and Breast Surgery, Zealand University Hospital, Roskilde, Denmark. The follow-up period for complications was one year. Data was obtained digitally through Sundhedsplatformen by Epic to a prefabricated database in REDCap (Research Electronic Data Capture) by Vanderbilt University. The database includes three categories: demographics, surgery, and complications. All investigators were employed at Zealand University Hospital, Roskilde. All surgeries were performed by different plastic surgeons from resident to attending level at the department of Plastic Surgery. The university department is considered as a high-volume center for skin cancer treatment. Initial data collection included 1312 patients receiving five different surgical interventions: simple primary wound closure, secondary healing, flap reconstruction, FTSG, and split-thickness skin grafts. After reviewing patient records and removing duplicates and uncompleted records, the total number of patients was 1283. The aimed patient population presented in this study is a group of 607 patients undergoing autologous flap- and FTSG. Reconstructions using a combination of flap and FTSG were excluded.

Data were collected uniformly by clearly written definitions of measurements susceptible to interpretation, such as definitions of comorbidities and complications (Table 1). Instructions in data collection and transfer were given both in writing and orally to the data collectors. Due to sensitive data, the records are not made public. Data was transferred from REDCap to R-studio for analysis. Data were analyzed using logistic regression for combinations of categoric and numeric predictors with categoric outcome variables. The Chi-square test was used to analyze categoric outcome variables with categoric predictors. An odds ratio was performed to analyze the association between risk factors and complications. A *p*-value of 0.05 or less was considered statistically significant, and a *p*-value of less than 0.001 was highly significant.

Definition of data item. The table shows short definitions of data susceptible to interpretation.

## 3. Results

### 3.1. Patient Demographics

The investigated group consisted of 607 patients surgically treated for BCC or SCC of the face, including 304 (49.9%) reconstructive flap surgeries and 303 (50.1%) FTSG procedures. Mean age at the time of surgery was 73.4 years (SD = 10.0) for flap and 76.7 years (SD = 10.1) for FTSGs reconstruction. Women were predominantly reconstructed using flaps (52.6%), and men were predominantly reconstructed by means of FTSG (53.1%). Mean body mass index (BMI) for both groups amounted 26.7 ± 4.8 for flap and 26.2 ± 4.9 for FTSG reconstruction. Comorbidities and group demographics are summarized in Table 2.

The groups carried most demographics without statistical differences. However, a few factors differed significantly. Patients undergoing FTSGs were significantly older, 76.7 years (SD = 10.1) vs. flaps 73.4 years (SD = 10.0, *p* =< 0.001), had significantly higher numbers of other cancers, 72 (23.8%) vs. 43 (14.1%, *p* = 0.003), and a higher fraction received blood thinners 148 (49%) vs. 129 (42%, *p* = 0.043).

### 3.2. Facial Distribution of Surgery

The study divided the face into seven distinct regions: frontal, temporal, periorbital, nasal, zygomatic/buccal, periocular, and mental. Flaps were distributed as follows: 5 in the frontal region (1.6%), 2 in the temporal region (0.7%), 15 in the periorbital region (4.9%), 25 around the cheeks (zygomatic region and buccal region, 8.2%), 237 in the nasal region (78.0%), 17 in the periocular region (5.6%) and 3 in the mental region (1.0%, see Figure 1a). The distribution of FTSG by region was: 60 in the frontal region (19.8%), 60 in the temporal region (19.8%), 31 in the periorbital region (10.2%), 22 around the cheeks (zygomatic region and buccal region) (7.3%), 126 in the nasal region (41.6%), 3 in the periocular region (1.0%) and 1 in the mental region (0.3%, see Figure 1b).

The most frequent site of surgery was the nasal region, (*n* = 363, 59.8%), where the defects were reconstructed approximately two-thirds as often with flaps (*n* = 237, 65.3%) as compared to FTSGs (*n* = 126, 34.7,3%). In the frontal (*n* = 60, 19.8%) and temporal regions (*n* = 60, 19.8%), FTSGs were used predominantly compared to flap surgery.

### 3.3. Facial Flap Variability

The variety of flaps consists of 13 different types utilized at least more than three times. Flap types performed less than three times were pooled in the “unspecified” category Table 3.

The 13 flap types were pooled based on tissue movement in rotation flaps, transposition flaps, and advancement flaps. Flaps without specification were designated to the group “unspecified flaps” (Figure 2a). Transposition flaps were the most frequent (*n* = 209, 69%), followed by rotational (*n* = 47, 15%), unspecified (*n* = 33, 11%), and advancement flaps (*n* = 15, 5%).

The most frequently used facial flaps were nasolabial flaps (*n* = 90, 29.6%), followed by bilobed flaps (*n* = 58, 19%), others (*n* = 37, 12.2%), rhomboid flaps (*n* = 35, 11.5%), and nasofrontal flaps (*n* = 25, 8.2%). Other specific flap types represented less than 5% of the combined flap surgeries. The facial distribution of flap types is presented in Figure 2b.

### 3.4. Complications

#### 3.4.1. Flap Surgery and FTSG

Patients undergoing FTSGs had a significantly higher incidence of hematomas, (*n* = 32, 10.5%) compared to flaps (*n* = 13, 4.2%, *p* = 0.003). Necrosis occurred more often in the FTSG-group (*n* = 56, 18.5%) compared to local flaps (*n* = 11, 3.7%, *p* < 0.001). When subgrouping the degree of necrosis, edge necrosis was not significantly different between the groups *p* = 0.56), but partial necrosis (*p* < 0.001) and total necrosis (*p* < 0.001) were both highly significant, favoring flaps (see definition of complications in Table 1). Other complications investigated were not significantly different between groups (Table 4).

The FTSG group, presented in Table 4, received significantly more blood thinners than the flap group. A subgroup analysis of this patient group clarified that it was not the underlying cause of the increased hematomas of FTSGs as the fraction of hematomas among patients receiving blood thinners was 13.5% (*n* = 20) and 4% (*n* = 5) in flaps (*p* = 0.21) and therefore did not present with statistically more hematomas than for their entire group population. Nevertheless, the difference was still statistically significant, favoring flaps (*p* = 0.003). As hematoma under the local flap or skin graft can cause tension of the overlying tissue and, in the worst case, lead to necrosis [7], a correlation test was made to predict the causality of hematoma on necrosis. The analysis found that hematomas significantly increase the incidence of necrosis for both FTSGs and flaps (*p* < 0.001).

#### 3.4.2. Flap Type-Associated Complications

The complication profile of the three different flap fashions, namely, transposition, rotation, and advancement, was investigated, but none of the flap designs was related to particularly increased complications (see Table 5).

#### 3.4.3. Patient Age and Complications

Patients were grouped as ≤75 years (younger) and > 75 years (older). The younger group consisted of 320 individuals with a mean age of 66.8 ± 7.9 years. The older group consisted of 287 patients with a mean age of 82.4 ± 5.1 years. The older group had significantly lower BMI, 25.9 ± 7.7, than the older group, 27.1 ± 4.8 (*p* = 0.005); significantly more frequent hypertension, 66.2% vs. 44.9% (*p* < 0.001); significantly more other cancers, 22.5% vs. 15.0% (*p* = 0.022); significantly more heart diseases, 45.9% vs. 23.3% (*p* < 0.001); and received more blood thinners 57.7% vs. 33.4% (*p* < 0.001). In summary, the older group had significantly more comorbidities, but none of the investigated distinct complications was found to differ significantly, necrosis (*p* = 0.16), hematoma (*p* = 0.39), infection (*p* = 0.20), cellulitis (*p* = 1.00), wound dehiscence (*p* = 0.65). However, the odds ratio for major complications found that age significantly increase the risk of major complications for FTSGs, (OR =1.06 [CI 1.02, 1.11], *p* = 0.007) but not for local flaps (OR = 1.02 [CI 0.98, 1.06], *p* = 0.4). The risk of major complications increases with age, but the data did not provide a specific age cut off to define a high-risk group.

#### 3.4.4. Facial Region and Complications

The seven defined facial regions, as depicted in Figure 1, were analyzed according distribution of either partial necrosis, total necrosis, or hematoma (see Table 6). Surgeries in the mental region were too few to be included in the analysis.

The risk of partial necrosis, total necrosis, or hematoma was highest in the temporal region (35.5%), followed by frontal region (26.1%), zygomatic/buccal region (19.2%), nasal region (13.0%), perioral region (10.0%) and lastly the periorbital region (8.7%). In the three regions with the most major complications, FTSG was performed predominantly (78.7% of the reconstructions). A total of 61% of surgeries were performed with flaps for the three regions with the fewest complications.

#### 3.4.5. Complications Related to Patient Demographics and Comorbidities

Possible impact of patient demographics and comorbidities on major complications was investigated for flaps and FTSG (see Table 7). Comorbidity had no significant impact on the development of major complications in both reconstruction groups. For FTSG, male sex and age were associated with a significantly increased risk of major complications.

#### 3.4.6. Tumor Size

The median tumor size of NMSCs amounted 10 mm (IQR = 7.0–17.0 mm, Table 5). FTSGs were applied to reconstruct significantly larger tumors (14.0 mm, IQR = 9.0–20.0 mm) compared to flaps (9.0 mm, IQR = 6.0–13.0 mm, *p* < 0.001). Data revealed that surgeons preferred flaps to reconstruct smaller defects in regions with low tissue mobility, such as the nasal region. Flaps were rarely preferred in areas with a better possibility of tissue recruitment compared to FTSGs (Table 8).

Tumor diameter was analyzed for its impact on complications. At tumor diameters > 15 mm, the risk of hematoma significantly increases from 6.0% to 12.5% (*p* = 0.01) for both FTSG and flaps. At tumor diameter > 20 mm, the risk of hematoma increases even further to 17.9% (*p* < 0.001). The risk of any kind of necrosis increases from 9.8% to 20.2% at tumor diameters > 20 mm (*p* = 0.014). No significant differences were found with increased tumor size for other investigated complications.

#### 3.4.7. Level of Surgical Experience

The study investigates the outcome of being operated by board-certified plastic surgeons compared to a surgical trainee to uncover if the level of training may affect complication rates between FTSG and flaps. The analysis did not find any significant difference in any singular complication or the body encompassing major complications for FTSGs (*p* = 0.66) or flaps (*p* = 0.15). The tumor sizes of both groups (FTSGs and flaps) were equal for board-certified plastic surgeons and surgical trainees.

## 4. Discussion

This single-center study of 607 patients that received either local flaps or FTSG in order to restore skin cancer-related facial defects revealed that flaps were predominantly chosen to reconstruct defects in the nasal region, with a transpositional flap design most frequently used. In other facial regions, FTSGs were applied more often. The single-center study was conducted at a large high-volume center for plastic surgery, where all kinds of reconstructive procedures including advanced flap surgery were available at any time. At regions that tends to have better skin mobility and tissue recruitment, we argue that more significant defects that do not allow primary wound closure may be reconstructed with flaps instead of FTSGs, possibly lessening the patient’s risk of major complications according to the present results. Further, due to the support of subcutaneous tissue that provides better skin function and decreases the risk of aesthetically disturbing skin elevation changes, local flaps allow reconstructions with a good color match [7,8]. Supportively, a study by Schnabl et al., including over 1800 patients, found that patient-reported outcomes were significantly better for flaps than for FTSGs [9].

However, functional, aesthetical and quality-of-life-associated outcomes are not the main focus here. In the present study, we wanted to clearly figure out the risk for minor and major complications related to local flaps and FTSG used for facial reconstruction after skin cancer. According to the present data, the complication profile favors flaps over FTSGs in terms of major complications. From the perspective of the current literature, Leibovitch et al. investigated the rate of necrosis of FTSGs of the head, neck, genitals and upper limps in 2673 patients. They found that 3.7% had partial or total graft necrosis [10], though the study did not provide a definition of necrosis. However, it is approximately 10% less than in the present cohort and a cohort by Keh et al., where 13.6% of 128 FTSG reconstructions of the face developed graft necrosis of more than 60% reconstructed surface [11], resembling the data presented in this study. The most noticeable difference between the studies is the age difference in the patient populations. In the large study by Leibovitch, mean patient age was 65 years [10] compared to 75 years in the current study and 71 years in the study by Keh et al. [11]. None of the abovementioned studies investigated the impact of age on complications.

This study found that for FTSG (*n* = 303 patients), age significantly increased the risk of major complications, which may explain part of the difference between study complications. The patient-related factors found to increase the risk of major complications indicate that selecting patients for FTSG must be made with more caution than flaps.

The present data also highlights that men exhibited a 3.72 times increased risk for major complication compared to women. It is known that men tend to have higher risk of complications [12,13], which may explain why recommendations of postoperative bolus care may not have been met to the same degree for each gender resulting in increased complications. Although this hypothesis is highly speculative, it offers the local flap procedure more straight forward and less demanding postoperative care compared to FTSG.

The rate of partial or total necrosis in patients receiving local flaps was low (0.7%). Compared to a similar study by Rustemeyer et al., this appears to be very low [14]. One explanation might be that these procedures were solely performed in a plastic surgery high-volume center for facial reconstruction, providing skilled surgeons and an academic program for reconstruction. This might be a single-center bias.

The present findings do not favor any flap-type regarding complications.

FTSGs presented significantly more major complications in the studied cohort, among others [11,14] which caused additional or prolonged hospital visits and nursing hours [8] The wound dressing applied to FTSGs can also be challenging to manage by the patients or caretakers. On the other hand, certain flap surgeries demand multiple-step procedures, such as the median forehead flap (the Indian method). However, they represent a minor fraction of the total body of flaps used in the clinical praxis for facial reconstruction following skin cancer. The pre-operational preparation of flaps is considerable as each flap must be designed based on the precise tumor location [15], whereas FTSGs require less preparation and planning time. It requires plastic surgical experience to design and to choose the appropriate flap for at given or expected defect. Therefore, the organizational requirements and a certain case load are crucial to provide a facial reconstruction program that enables the best patient care and training by means of experienced reconstructive senior surgeons. The complication rates were equal when comparing less experienced and skilled surgeons in the present study. Similar findings have previously been reported in NMSC surgery of the face.

Hematomas significantly increased the risk of necrosis for FTSGs and flaps, especially for larger tumors. A fraction of minor hematomas may be underreported after flap reconstruction as it might be clinically invisible due to coverage of the “thicker” flap and can usually be handled with elevation and pressure. Patients receiving FTSGs are potentially more predisposed to hematoma-related reconstructive failure, as FTSGs demand diffusion to prevent necrosis and to enable take and healing. Thus, a hematoma layer can directly act as a diffusion-barrier [7].

Patients undergoing FTSG reconstruction always demand a post-operational control with bolus unwrapping and clinical evaluation.

There are several limitations of the present study. First, it is a single-center study that uses retrospective data. Second, the study did not assess the prognostic, aesthetic, functional or quality-of-life outcome after local flap and FTSG reconstruction, limiting the overall evaluation of the surgeries [9]. Our research group has initiated a prospective study to investigate patient-reported satisfaction and quality of life measurements to gather information on the patient’s surgical journey and self-assessed outcome. By means of this prospective approach, we expect to gain a better and fuller picture of the local flaps and FTSGs in facial reconstruction after skin cancer. Other studies investigating patient-reported outcomes after facial reconstruction suggest that flap reconstructions are associated with superior patient-reported aesthetic outcomes [8,14,16,17].

In conclusion, we still believe that reconstructive planning in patients suffering from skin cancer that demands reconstruction should be a highly individualized process that addresses the individual patient need and wishes in terms of personalized medicine. Data from 607 patients revealed that both local flaps and FTSGs are safe procedures that can be offered to patients with demand for facial reconstruction after cancer removal. Patients undergoing FTSGs for facial reconstruction have a higher risk of hematoma (*p* = 0.003) and partial- (*p* < 0.001) or total (*p* < 0.001) necrosis compared to flap reconstructions. With regard to risk stratification, the present study indicates that clinical recommendations for FTSGs should be made with caution especially in elderly patients and/or male patients due to a significant and distinctive higher risk for major complications. As expected, large tumors should be considered at high risk of hematoma and necrosis after reconstruction.

## Figures and Tables

**Figure 1 jpm-12-02067-f001:**
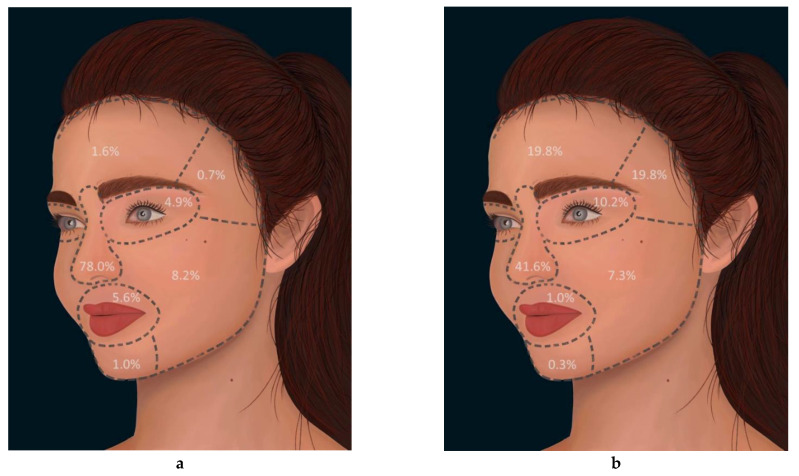
Distribution of flaps and full thickness skin graft (FTSG) according to distinct facial units. (**a**) Distribution of flaps according to the facial region. (**b**) Distribution of FTSG by the facial regions.

**Figure 2 jpm-12-02067-f002:**
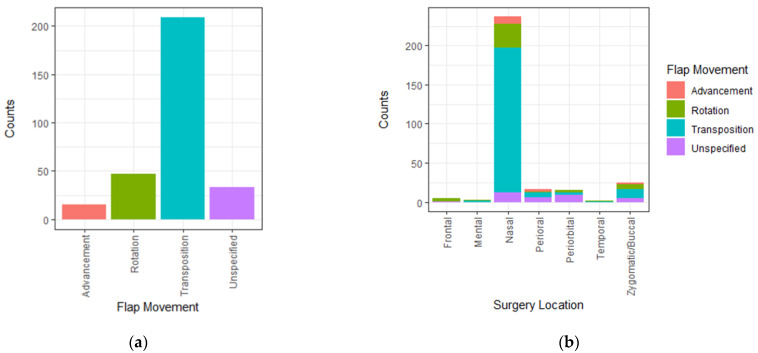
Distribution of flap types and facial regions. (**a**) Distribution of flap types and (**b**) flap design used according to distinct facial region.

**Table 1 jpm-12-02067-t001:** Definition of terminology.

Term	Definition of the Collected Data
BMI	Body mass index measured ≤30 days before surgery.
Autoimmune disease	Has an autoimmune diagnosis code in their medical journal. Including the following: - Inflammatory bowel disease - Musculoskeletal autoimmune disease - Skin autoimmune disease - Infectiously initiated autoimmune disease - Autoimmune organ failure - Combinatory autoimmune diseases
Hypertension	Has the diagnosis code in their medical journal both well-regulated and unregulated? Including the following: - White coat hypertension - Hypertension of unknown origin - Hypertensio arterialis essentialis - Hypertention secundary to other disease - Hypertensio renovascularis - Hypertensio arterialis by other renal diseases - Hypertensio arterialis by endocrine diseases
Other cancer	Has a cancer diagnosis or previous cancer diagnosis, and still in follow-up, other than NMSC.
Heart disease	Has a heart disease diagnosis code in their medical journal, both well treated and untreated. Including following: - Ischemic heart diseases - Acute coronary syndrome - Acute myocardial infarction - Chronic ischemic heart diseases - Pulmonary heart diseases - Heart valve disease - Heart failure - Atherosclerosis
Blood thinners	Take blood thinners regularly. The patients are advised to pause their medication prior to surgery. However, comorbidities may have led to the patients continuing due to the risk of pause.
Alcoholic drink	One substance contains 8 g of alcohol.
Previous smoker	Quit smoking ≥30 days before the surgery.
Infection	Clinically observed and required antibiotic treatment.
Cellulitis	Clinically observed.
Minor hematoma	No surgical revision required.
Major hematoma	Required surgical revision.
Edge necrosis	≤5% of the transplant or flap.
Partial necrosis	>5% to 66% of the transplant or flap.
Total necrosis	>66.6% of the transplant or flap.
Major complication	Hematoma, partial necrosis, total necrosis, wound dehiscence, and infection.

**Table 2 jpm-12-02067-t002:** Patient demographics and comorbidities.

	FTSG (*n*)	Flap (*n*)	*p*-Value
*n*	303	304	
Age, (mean [^a^ SD])	76.7 (10.1)	73.4 (10.0)	<0.001
Sex = Male (%)	161 (53.1)	144 (47.4)	0.168
BMI (mean [^a^ SD])	26.24 (4.9)	26.74 (4.8)	0.263
Autoimmune disease (%)	31 (10.2)	27 (8.9)	0.584
Hypertension (%)	171 (56.4)	170 (55.9)	0.935
Other cancer (%)	72 (23.8)	43 (14.1)	0.003
Diabetes (%)	40 (13.2)	42 (13.8)	0.906
Heart condition (%)	107 (35.3)	107 (35.2)	1.000
Blood thinners (%)			0.043
No	155 (51.2)	178 (58.6)	
Acetylic Acid	48 (15.8)	29 (9.5)	
Other blood thinners	100 (33.0)	97 (31.9)	
^b^ Alcohol (%)			0.736
<7 units per week	142 (46.9)	154 (50.7)	
7–14 units per week	43 (14.2)	41 (13.5)	
>14 units per week	16 (5.3)	18 (5.9)	
Not reported	102 (33.7)	91 (29.9)	
Smoking (%)			0.950
No	127 (41.9)	124 (40.8)	
Active smoker	40 (13.2)	39 (12.8)	
Former smoker	56 (18.5)	62 (20.4)	
Not reported	80 (26.4)	79 (26.0)	

^a^ SD = standard deviation, ^b^ alcohol, *n* = one drink containing 8 g of alcohol.

**Table 3 jpm-12-02067-t003:** Variety of utilized flaps.

Flap Type	Number = *n*
Nasolabial	90
Hatchet	5
Frontonasal	25
Unspecified transposition	6
Paramedian Forehead	9
Random	3
Rhomboid	35
Unspecified rotation	14
Shark island	3
Spear	3
Bilobed	58
Trilobed	4
V-Y	12
Unspecified	37
**Total**	**304**

**Table 4 jpm-12-02067-t004:** Complications related to FTSGs and local flaps.

Complication	FTSG	Flaps	*p*-Value
Infection (%)	25 (8.3)	28 (9.2)	0.774
Cellulitis (%)	7 (2.3)	5 (1.6)	0.577
Hematoma (%)			0.003
No	271 (89.4)	291 (95.7)	
Minor hematoma	31 (10.2)	12 (3.9)	
Major hematoma	1 (0.3)	1 (0.3)	
Wound dehiscence (%)	28 (9.2)	20 (6.6)	0.233
Necrosis (%)			<0.001
No	247 (81.5)	293 (96.4)	
Edge necrosis	11 (3.6)	9 (3.0)	
Partial necrosis	22 (7.3)	2 (0.7)	
Total necrosis	23 (7.6)	0 (0.0)	

**Table 5 jpm-12-02067-t005:** Complications related to principle flap design.

	Advancement	Rotation	Transposition	Unspecified	*p*-Value
*n*	15	47	209	33	
Infection (%)	1 (6.7)	4 (8.5)	20 (9.6)	3 (9.1)	1.000
Cellulitis (%)	0 (0.0)	0 (0.0)	3 (1.4)	2 (6.1)	0.259
Hematoma (%)					0.436
No	15 (100.0)	45 (95.7)	200 (95.7)	31 (93.9)	
Minor hematoma	0 (0.0)	2 (4.3)	9 (4.3)	1 (3.0)	
Major hematoma	0 (0.0)	0 (0.0)	0 (0.0)	1 (3.0)	
Wound dehiscence (%)	1 (6.7)	2 (4.3)	15 (7.2)	2 (6.1)	0.971
Necrosis (%)					0.893
No	15 (100.0)	45 (95.7)	200 (95.7)	33 (100.0)	
Edge necrosis	0 (0.0)	2 (4.3)	7 (3.3)	0 (0.0)	
Partial necrosis	0 (0.0)	0 (0.0)	2 (1.0)	0 (0.0)	
Total necrosis	0 (0.0)	0 (0.0)	0 (0.0)	0 (0.0)	

**Table 6 jpm-12-02067-t006:** Partial necrosis, total necrosis, or hematoma by facial region.

Region	Frontal	Mental	Nasal	Perioral	Periorbital	Temporal	Zygomatic/Buccal
*n*	65	-	363	20	46	62	47
Risk of partial necrosis, total necrosis, or hematoma	26.1%	-	13%	10.0%	8.7%	35.5%	19.2%

**Table 7 jpm-12-02067-t007:** Impact of patient demographics on major complications.

Characteristic	Flaps			FTSG		
	OR ^1^	95% CI ^1^	*p*-Value	OR ^1^	95% CI ^1^	*p*-Value
Age	1.02	0.98, 1.06	0.4	01.06	1.02, 1.11	0.007
Sex						
Female	—	—		—	—	
Male	1.33	0.60, 2.96	0.5	3.72	1.72, 8.43	0.001
BMI	0.93	0.84, 1.02	0.12	0.96	0.88, 1.04	0.4
Autoimmune disease						
No	—	—		—	—	
Yes	2.13	0.57, 7.20	0.2	1.90	0.60, 5.91	0.3
Hypertension						
No	—	—		—	—	
Yes	0.57	0.24, 1.37	0.2	1.92	0.80, 4.82	0.2
Other cancer						
No	—	—		—	—	
Yes	0.39	0.08, 1.42	0.2	0.87	0.36, 2.01	0.7
Diabetes						
No	—	—		—	—	
Yes	2.38	0.80, 6.86	0.11	0.54	0.18, 1.54	0.3
Heart disease						
No	—	—		—	—	
Yes	2.67	0.93, 7.97	0.071	0.59	0.23, 1.46	0.3
Blood thinners						
ASA	—	—		—	—	
Others	0.46	0.11, 2.19	0.3	0.80	0.26, 2.43	0.7
No	0.74	0.18, 3.58	0.7	0.61	0.18, 2.05	0.4

^1^ OR = Odds Ratio, CI = Confidence Interval.

**Table 8 jpm-12-02067-t008:** Tumor size.

	Flaps	FTSG	Total
Size of the tumors, mm (median, [IQR])	9.0 [6.0, 13.0]	14.0 [9.0, 20.0]	10.0 [7.0, 17.0]

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
