# Peer review of "Risk Stratification of Local Flaps and Skin Grafting in Skin Cancer-Related Facial Reconstruction: A Retrospective Single-Center Study of 607 Patients"

_jpm, 2022, doi:10.3390/jpm12122067_

Round 1

Reviewer 1 Report

The paper is well structured, fits well the journal scope and present nice conclusions. In my opinion it is ready for publication

Author Response

Thanks. No revisions have been made for reviewer 1.

Reviewer 2 Report

This is a very well-designed retrospective study, with a large amount of interesting data that will undoubtedly be of interest to the scientific population interested both in the field of clinical and translational dermatology and in the field of advanced therapies and personalized medicine. This paper fits very well with the special issue offered by this journal.

Some minor revisions are as follows:

Line 72 to line 76. In this paragraph I recommend adding a flow chart indicating the initial data collection and the final number of patients included in the study. With arrows indicating that duplicate cases or cases different from the surgeries analyzed were eliminated. This would be a simple and visual way to explain this paragraph.

Please specify in Table 1 definitions of the collected data about  Autoimmune disease, Hypertension and Heart disease.

In the text of the manuscript I have not located the origin of either the flaps or the FTSGs. Could you indicate FTSGs types that have been used in the surgeries in Matherial and Methods section?

One issue that has not been addressed is the health care costs, which is also essential when choosing one type of surgery over another. Is there a significant difference in the health care costs involved in one method or the other? In this case, which method implies a lower health care cost and why?

There were no autoimmune rejections of the transplants as a post-operative complication in any patient?

Would it be possible to do reconstructive surgery with flap and FLSGs combined?

The study concludes that patients undergoing FTSGs for reconstruction have a significantly higher risk of hematoma and necrosis than flap reconstructions. Is it possible that this is because the wounds chosen for FTSG surgery had greater depth and surface area?

As you comment in the text, patient satisfaction after reconstructive surgery is one of the four pillars of reconstructive oncology. Therefore, I would recommend to have added in this paper a prospective study to investigate patient-reported measures of satisfaction and quality of life questionaires. If you have not yet performed these analyses, you may publish them soon. since the aesthetic outcome of flaps and FTSGs as well as the quality of life of the operated patients is also essential in the area of reconstructive surgery.

The main limitation of this paper is that the references used are adequate but scarce, there is not much background on reconstructive surgery with artificial skin such as FLSGs, something that has been investigated for some time, but the use of this type of skin designed by tissue engineering in clinical practice is very new and current. These papers may be of interest to improve the content background of the manuscript. doi: 10.3390/ijms21218197. doi: 10.1016/j.transproceed.2019.08.014 doi: 10.1016/j.transproceed.2019.08.014

Author Response

This is a very well-designed retrospective study, with a large amount of interesting data that will undoubtedly be of interest to the scientific population interested both in the field of clinical and translational dermatology and in the field of advanced therapies and personalized medicine. This paper fits very well with the special issue offered by this journal. Some minor revisions are as follows:

Line 72 to line 76. In this paragraph I recommend adding a flow chart indicating the initial data collection and the final number of patients included in the study. With arrows indicating that duplicate cases or cases different from the surgeries analyzed were eliminated. This would be a simple and visual way to explain this paragraph.

Answer: Thanks for the comment. A flow chart has not been added to the revised paper. The article states the final number of patients collected and included in the study. We believe it’s done in a simple and easily understandable way. If the editor strongly desires a visualized version of the text, we would of course happily provide such a figure.

P2 l83-89

 “The university department is considered as a high-volume center for skin cancer treatment. Initial data collection included 1,312 patients receiving five different surgical interventions: simple primary wound closure, secondary healing, flap reconstruction, FTSG, and split-thickness skin grafts. After reviewing patient records and removing duplicates and uncompleted records, the total number of patients was 1.283. The aimed patient population presented in this study is a group of 607 patients undergoing flap- and FTSG.”

Please specify in Table 1 definitions of the collected data about Autoimmune disease, Hypertension and Heart disease.

Answer: Thanks for the comment. Table 1 states following

“Has an autoimmune diagnosis code in their medical journal”

Has been changed to include the diagnostic codes

“Has an autoimmune diagnosis code in their medical journal.

Including the following: 

  • Inflammatory bowel disease
  • Musculoskeletal autoimmune disease
  • Skin autoimmune disease
  • Infectiously initiated autoimmune disease
  • Autoimmune organ failure
  • Combinatory autoimmune diseases”

Has the diagnosis code in their medical journal both well-regulated and unregulated?

Answer: Has been changed to add the diagnostic codes included.

“Has the diagnosis code in their medical journal both well-regulated and unregulated?

Including the following:

  • White Coat hypertention
  • Hypertension of unknown origin
  • Hypertensio arterialis essentialis
  • Hypertention secundary to other disease
  • Hypertensio renovascularis
  • Hypertensio arterialis by other renal diseases
  • Hypertensio arterialis by endocrine diseases”

“Has a heart disease diagnosis code in their medical journal, both well treated and untreated”

Has been changed to include the diagnostic codes

“Has a heart disease diagnosis code in their medical journal, both well treated and untreated.

Including following:

  • Ischemic heart diseases
  • Acute coronary syndrome
  • Acute myocardial infarction
  • Chronic ischemic heart diseases
  • Pulmonary heart diseases

  • Heart valve disease

  • Heart failure

  • Atherosclerosis”

In the text of the manuscript I have not located the origin of either the flaps or the FTSGs. Could you indicate FTSGs types that have been used in the surgeries in Matherial and Methods section?

Answer: thanks for the comment. The word “facial” has been added to the sentence in material and methods:

P2 – L75 “Data was collected between 01.12.2021 to 25.04.2022 and included patients undergoing facial surgery for BCC and SCC…

One issue that has not been addressed is the health care costs, which is also essential when choosing one type of surgery over another. Is there a significant difference in the health care costs involved in one method or the other? In this case, which method implies a lower health care cost and why?

Answer: Thanks for the comment. This has not been investigated in the current study although relevant for future decision-making. The aim of this study was to evaluate the risk of complications therefore I do not believe that the cost of the procedure fits the scope of the article. In the following section of the study

P13 – l312-318 it says

FTSGs presented significantly more major complications in the studied cohort, among others [11,14] which caused additional or prolonged hospital visits and nursing hours [8] The wound dressing applied to FTSGs can also be challenging to manage by the patients or caretakers. On the other hand, certain flap surgeries demand multiple-step procedures, such as the median forehead flap (the Indian method). They, however, represent a minor fraction of the total body of flaps used in the clinical praxis for facial reconstruction following skin cancer.”

Here we discuss the pros and cons in terms of time spend from different healthcare personnel.

There were no autoimmune rejections of the transplants as a post-operative complication in any patient?

Answer: Thanks for the comment. Local flap surgery relies on local tissue and will always be autologous. Full thickness skin grafts are in these cases also always autologous as allogenic/xenogenic skin grafts are rarely performed outside extensive burns.  

For clarification the word “Autologous” has been added in the materials / method section

P2 – I88“ The aimed patient population presented in this study is a group of 607 patients undergoing autologous flap- and FTSG...”

Would it be possible to do reconstructive surgery with flap and FLSGs combined?

Answer: Thanks for the comment. Yes, for larger defects it could be reconstructed with a FTSG and adjusted with a smaller flap or vise versa. Those types of combinations are however rare compared to the stand-alone procedures and were therefore not included in the study.

Following has been added to the materials section

P2 – l98-90“Reconstructions using a combination of flap and FTSG were excluded.”

The study concludes that patients undergoing FTSGs for reconstruction have a significantly higher risk of hematoma and necrosis than flap reconstructions. Is it possible that this is because the wounds chosen for FTSG surgery had greater depth and surface area?

Answer: Thanks for the comment. The depth of the wounds after excision of the tumours was not collected for the study. As the skin and subcutaneous tissue of the face has a rather low thickness most tumours are excised to more dense structures such as cartilage or bone. The thickness of the wound was therefore not a parameter of interest.     

As you comment in the text, patient satisfaction after reconstructive surgery is one of the four pillars of reconstructive oncology. Therefore, I would recommend to have added in this paper a prospective study to investigate patient-reported measures of satisfaction and quality of life questionaires. If you have not yet performed these analyses, you may publish them soon. since the 2 aesthetic outcome of flaps and FTSGs as well as the quality of life of the operated patients is also essential in the area of reconstructive surgery.

Answer: Thanks for the comment. As this is a retrospective study it was not possible to collect adequate information from the patients. We have however initiated a study as described in the discussion:

Our research group has initiated a prospective study to investigate patient-reported satisfaction and quality of life measurements to gather information on the patient’s surgical journey and self-assessed outcome. By means of this prospective approach we expect to gain a better and fuller picture of the local flaps and FTSGs in facial reconstruction after skin cancer.”

The main limitation of this paper is that the references used are adequate but scarce, there is not much background on reconstructive surgery with artificial skin such as FLSGs, something that has been investigated for some time, but the use of this type of skin designed by tissue engineering in clinical practice is very new and current. These papers may be of interest to improve the content background of the manuscript. doi: 10.3390/ijms21218197. doi: 10.1016/j.transproceed.2019.08.014 doi: 10.1016/j.transproceed.2019.08.014

Answer: Thanks for the comment. The exiting subject of tissue scaffolding, autologous stem cells culture expanding of cells and ECM and the immense possibility of expanding the canvas of skin (allogenic or autologous) in innovative ways is truly exiting. I (the first author) have worked with autologous ex-vivo expanded adipose-derived stem/stromal cells and extracellular matrixes for wound healing, skin age-reversing and fat grafting for five years. I am currently drafting my PhD thesis in this field. However, the exciting potential of these new tissue engineering, with all respect, I think will confuse the readers more than it will benefit, as it is not investigated in the current study. At the current state of the tissue engineering its relevance does not lie with small 10-14 mm wound after facial cancer but with large burns as it is also presented in the three papers that the reviewer has referred to.          

Reviewer 3 Report

Excellent study. Please Structure the Abstract in Sections.

best regards

Author Response

Thanks the for comments. The abstract has been changed in the second revision.